# Intra- and Inter-Cellular Awareness for 3D Neuron Tracking and Segmentation in Large-Scale Connectomics

**Hao Zhai**[1,2]                                                                    ZHAIHAO2020@IA.AC.CN
**Jing Liu**[3]                                                                        LIUJING2016@IA.AC.CN
**Bei Hong**[4]                                                                       HONGBEI@CPL.AC.CN
**Jiazheng Liu**[1,2]                                                          LIUJIAZHENG2018@IA.AC.CN
**Qiwei Xie**[5]                                                                    QIWEI.XIE@BJUT.EDU.CN
**Hua Han**[1,2,3]                                                                     HUA.HAN@IA.AC.CN

[1] *State Key Laboratory of Multimodal Artificial Intelligence Systems,*
*Institute of Automation, Chinese Academy of Sciences*
[2] *School of Future Technology, University of the Chinese Academy of Sciences*
[3] *Research Center for Brain-inspired Intelligence,*
*Institute of Automation, Chinese Academy of Sciences*
[4] *Changping Laboratory*
[5] *Research Base of Beijing Modern Manufacturing Development, Beijing University of Technology*

**Editors:** Accepted for publication at MIDL 2023

## Abstract

Currently, most state-of-the-art pipelines for 3D micro-connectomic reconstruction deal with neuron over-segmentation, agglomeration and subcellular compartment (nuclei, mitochondria, synapses, *etc.*) detection separately. Inspired by the proofreading consensus of experts, we established a paradigm to acquire priori knowledge of cellular characteristics and ultrastructures, as well as determine the connectivity of neural circuits simultaneously. Following this novel paradigm, we were keen to bring the **I**ntra- and **I**nter-**C**ellular **A**wareness back when **T**racking and **S**egmenting neurons in connectomics. Our proposed method (II-CATS) utilizes *few-shot learning* techniques to encode the internal neurite representation and its learnable components, which could significantly impact neuron tracings. We further go beyond the original **e**xpected **r**un **l**ength (ERL) metric by focusing on **b**iological constraints (bERL) or spanning from the **n**ucleus to spines (nERL). With the evaluation of these metrics, we perform typical experiments on multiple electron microscopy datasets on diverse animals and scales. In particular, our proposed method is naturally suitable for tracking neurons that have been identified by staining.

**Keywords:** connectomics, neuron tracking, neuron segmentation, few-shot learning

## 1. Introduction

Connectomics, which reconstructs the synaptic level connectivity between neurons in electron microscopy, is paving the way toward learning the structure of the brain. With the advancements in high-resolution and high-throughput imaging with various electron microscopy, every interested cellular boundary and subcellular organelle (nuclei, mitochondria, *etc.*) could be clearly visible. However, to achieve the above, the datasets of brain tissue may now exceed a *peta*byte in size (Shapson-Coe et al., 2021; Bae et al., 2021), for which we need feasible automatic object tracking and segmentation methods.

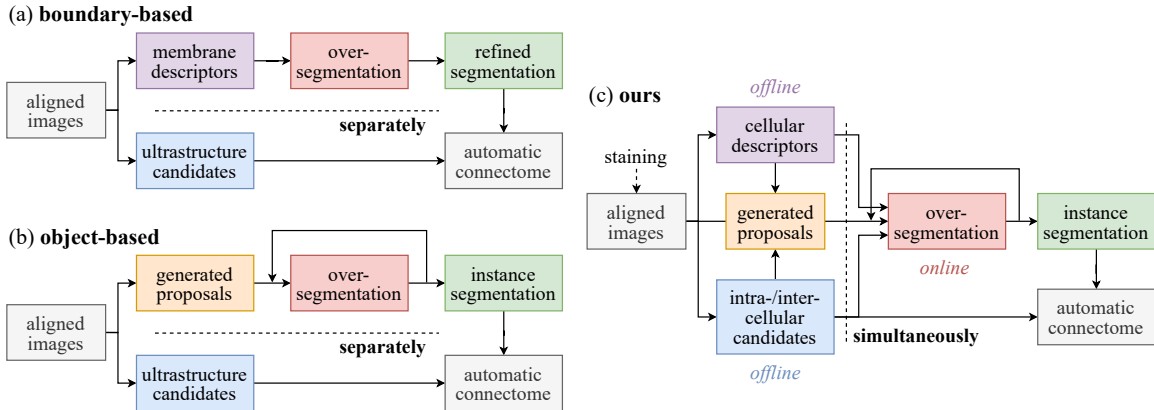

Figure 1: Comparison of current paradigms and our novel paradigm. Both **(a)** and **(b)** treat cellular and subcellular structures separately. In contrast, **(c)** obtains this information simultaneously.

Current state-of-the-art high-precision neuron reconstruction pipelines have already been able to fully automatically handle the cubic $10\mu m$ scale *saturate* reconstruction (Lee et al., 2017) and cubic $100\mu m$ *dense* reconstruction (Januszewski et al., 2018). These results are evaluated by voxel-wise membrane **boundaries** and instance-wise neuron **objects**, respectively. Due to the huge computation and labor cost, the large-scale *dense* connectome and *sparse* projectome reconstruction of imperfect datasets still remains challenging (Ngai, 2022). Some **boundary-based** approaches (Bailoni et al., 2020; Lee et al., 2021; Sheridan et al., 2022) use U-Nets (Ronneberger et al., 2015; Çiçek et al., 2016) flexibly, predicting intermediate descriptors for segmentation and running the graph-based agglomeration, as depicted in Figure 1(a). Some **object-based** approaches (Januszewski et al., 2018; Meirovitch et al., 2019; Gonda et al., 2021) execute single-neuron filling iteratively, extending the tracing area from seed points, as depicted in Figure 1(b). Other **skeleton-based** approaches (Motta et al., 2019; Schmidt et al., 2022) only focus on tracking the neuron skeletons for further connectomic analysis. Both (a) and (b) treat cellular and subcellular structures separately, lacking the procedure of integrating extra biological information.

Aside from these two pipelines, we consider introducing a new paradigm that could take almost full advantage of both, as depicted in Figure 1(c). With the consensus in micro-connectomics (Lee et al., 2019), the *voxel*-level accuracy of cellular descriptors (*e.g.*, boundaries, affinities, and shape descriptors) and subcellular candidates (*e.g.*, mitochondria and synaptic clefts) usually exceeds *instance*-level neuron reconstruction. Therefore in our paradigm, these more reliable offline results can actually be put to good use simultaneously as a priori for a more robust online procedure from neurite proposals to segments.

Following the novel paradigm, we proposed a method to absorb these various masks by integrating a differentiable few-shot learner into the encoder-decoder architecture, which *learns what to learn* (Bhat et al., 2020). In other words, our method enables the decoder to learn the rich internal learnable representation of object targets and region weights in masks (three small patches in Figure 2) instead of directly assembling from raw masks.

We further determine what the few-shot learner should learn around the structure of neurons. To generate primary proposals $y^b$, we build a *quasi-dense* (Pang et al., 2021)

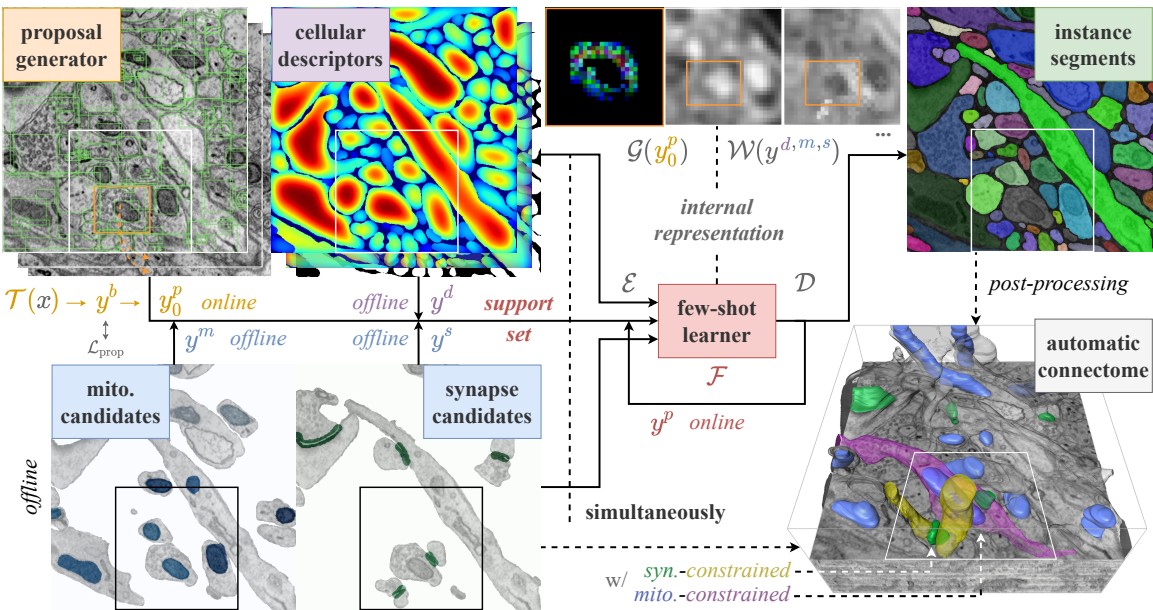

Figure 2: Implementation of our paradigm in Figure 1(c). The few-shot learner $\mathcal{F}$ in encoder-decoder $\mathcal{E}$-$\mathcal{D}$ architecture acquires offline priori to constrain online **p**redicted segments $y^p$. The quasi-dense tracker $\mathcal{T}$ generates **b**ox proposals $y^b$ from serial section images $x$. These further produced $y_0^p$, along with cellular **d**escriptors $y^d$, **m**itochondrion and **s**ynapse candidates $y^{m,s}$, contribute to the support set for $\mathcal{F}$. **Small patches:** three samples of internal representation from $\mathcal{G}$ and $\mathcal{W}$.

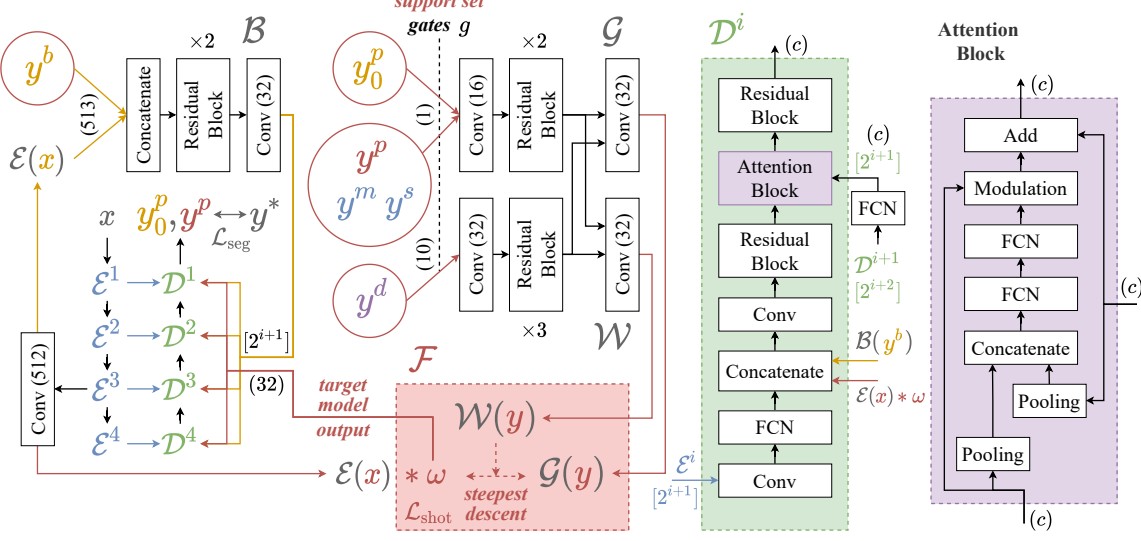

Figure 3: Details of our architecture in Figure 2. The third encoder block features $\mathcal{E}(x)$ are employed for matching with priori masks $y$ in the support set (red circles) by two pathways: decoding box representation $\mathcal{B}(y^b)$ into initial prediction $y_0^p$ (yellow arrows), or decoding target model output $\mathcal{E}(x) * \omega$ into subsequent prediction $y^p$ (red arrows). **Red:** the few-shot learner optimizes $\mathcal{E}(x) * \omega$ to the targeted $\mathcal{G}(y)$ with weighted $\mathcal{W}(y)$ online (dashed arrows). **Green:** the decoder block. **Purple:** the channel attention block integrates same-level target matching (bottom) and higher-level decoding (right) inputs. $y^*$, ground truth; [·], feature strides; (·), output channels.

tracker $\mathcal{T}(x)$ for *box*-level matching on local consistency across sections $x$. To make use of multifarious characteristics, we implemented emerging works (Sheridan et al., 2022) for *voxel*-level cellular descriptors $y^d$ and *pixel*-level subcellular candidates $y^{m,s}$ (Figure 2 left).

We perform experiments on a mouse dataset Kasthuri (Kasthuri et al., 2015), a songbird dataset J0126 (Januszewski et al., 2018), and a stained dataset (Joesch et al., 2016) by *voxel*-level metrics VI (Meilă, 2003) and *skeleton*-level metric expected run length (ERL). Inspired by the proofreading consensus of experts, we extended the original ERL metric by focusing on biological constraints (bERL) or spanning from the nucleus to spines (nERL). Our method sets new state-of-the-art of ERL-based metrics on certain datasets compared to other competitive neuron tracking methods in the field of connectomics.

## 2. Methodology

Our proposed method (II-CATS) utilizes *few-shot learning* for implementation of the novel paradigm in Figure 1(c). We illustrated some samples of input, output, and intermediate results of our pipeline in Figure 2. We also deconstructed the details of our encoder-decoder $\mathcal{E}$-$\mathcal{D}$ architecture in Figure 3. For simplicity, $x$ is an image from serial sections of the brain tissue, $y$ is the corresponding segmentation mask of certain biological structures, $\mathcal{E}(x)$ is defined as $\mathcal{E}^3\left(\mathcal{E}^2\left(\mathcal{E}^1(x)\right)\right)$, and $\mathcal{D}(\mathcal{E}(x), \cdot)$ is an abbreviation of $\mathcal{D}\left(\mathcal{E}^1(x), \mathcal{E}^2(x), \mathcal{E}^3(x), \mathcal{E}^4(x), \cdot\right)$.

The integral pipeline starts with **b**ox proposals $y^b$ of targets, which are generated from the quasi-dense tracker $\mathcal{T}$. Then, these proposals get through the box representation extractor $\mathcal{B}$ and segmentation decoder $\mathcal{D}$, producing the initial **p**redicted segments $y_0^p = \mathcal{D}\left(\mathcal{E}(x_0), \mathcal{B}\left(\mathcal{E}(x_0), y^b\right)\right)$. Next, these neurite segments, along with cellular **d**escriptors $y^d$, **m**itochondrion and **s**ynapse candidates $y^{m,s}$, contribute to the support set $\{(x, y)\}$ (Figure 2 left, the cellular and subcellular candidates could be precomputed offline). After that, the few-shot learner $\mathcal{F}$ acquires the internal representation from this support set (Figure 2 right, the initial proposals and recurrent predictions should be tracked online). The target model output $\mathcal{E}(x_s) * \omega_s$ is optimized to the targeted $\mathcal{G}(y_s)$ with weighted $\mathcal{W}(y_s)$. And the learned target filters $\omega_s$ are used for producing the subsequent **p**redicted segments $y_{s+1}^p = \mathcal{D}\left(\mathcal{E}(x_{s+1}), \mathcal{E}(x_{s+1}) * \omega_s\right)$ section by section $(\cdot \to s \to s+1 \to \cdot)$.

All modules in II-CATS are open source[1] and still under continuous development. Alternative details about the modules described below are provided in Appendix C.

### 2.1. Quasi-Dense Tracker ($\mathcal{T}$)

We take the popular region proposal network (RPN) with feature pyramid networks as the base of our tracker (Ren et al., 2015). The RPN generates regions of interest (*i.e.*, anchors) for the detection head with classification and regression branches. These was trained by three corresponding losses $\mathcal{L}_{\text{box}} = \mathcal{L}_{\text{rpn}} + \gamma_1 \mathcal{L}_{\text{cls}} + \gamma_2 \mathcal{L}_{\text{reg}}$, and $\gamma_1, \gamma_2$ are the loss weights.

On this basis, we merge the plenty of anchors to reasonable proposals by learning instance similarity across sections with *quasi-dense* matching. The *quasi-dense* matching only considers the potential neuron candidates at informative regions (Pang et al., 2021), which is better than *dense* (high computation intensity) or *sparse* (less training pairs) for crowded

---

1. https://github.com/JackieZhai/II-CATS

neurites. As for the architecture after RPN, an extra lightweight embedding head is added in parallel with the detection head. This was trained by contrastive loss $\mathcal{L}_{\text{embed}}$ (17).

Overall, the quasi-dense tracker $\mathcal{T}$ was trained by $\mathcal{L}_{\text{prop}} = \mathcal{L}_{\text{box}} + \mathcal{L}_{\text{embed}}$, and it generates neurite proposals $y^b$ by matching anchor candidates (yellow arrows in Figure 2 top-left).

### 2.2. Internal Representation Extractor $(\mathcal{B}, \mathcal{G}, \mathcal{W})$

A variety of cutting-edge few-shot learning strategies have been implemented for volume-EM images due to the special characteristics of certain biological structures. Inspired from (Bhat et al., 2020), we apply the *learning what to learn* strategy for a glance at a few masks $y$.

As shown in Figure 3 bottom-left, the two pathways of the $\mathcal{D}(\mathcal{E}(x), \cdot)$ decode either box proposals of neurites or segment masks of cellular, intra- and inter-cellular structures. Predicting only from single- or several-channel masks severely limits the information passed to the decoder $\mathcal{D}$, and thus increases the training burden of the $\mathcal{D}_\theta(\mathcal{E}_\theta(x), \cdot)$. To address this problem, we expanded these masks to multi-channel representation, learning what our internal few-shot learner $\mathcal{F}$ and subsequent decoder $\mathcal{D}$ should learn.

Along the box pathway (yellow arrows in Figure 3), we introduced the box representation extractor $\mathcal{B}$. It takes $y^b$ along with backbone features $\mathcal{E}(x)$ as input and predicts mask-related encoding. Along the segment pathway (red arrows in Figure 3), we introduced the target and weight representation extractor $\mathcal{G}$ and $\mathcal{W}$. $\mathcal{G}(y)$ sets multi-dimensional targets for target model outputs $\mathcal{E}(x) * \omega$, and $\mathcal{W}(y)$ tunes corresponding weights of different objects (big or small) and regions (clear or ambiguous). All extractors in use have 32 channels.

### 2.3. Few-Shot Learner $(\mathcal{F})$

Above all, we designed the target model for mapping the backbone features $\mathcal{E}(x)$ to the internal representation of masks $y$. It is simply implemented as the convolution $\mathcal{E}(x) * \omega$, and the few-shot learner $\mathcal{F}$ learns the target filter $\omega$. Then, we push forward our few-shot learner to minimize the squared error between the target model output and the various internal representations $\mathcal{G}(y)$, which are weighted by the element-wise attention $\mathcal{W}(y)$:

$$\mathcal{L}_{\text{shot}}(\omega) = \frac{1}{2} \sum_{\{(x,y)\}} \|\mathcal{W}(y) \circ (\mathcal{E}(x) * \omega - \mathcal{G}(y))\|^2 + \frac{\lambda}{2} \|\omega\|^2, \tag{1}$$

where the $\{(x, y)\}$ is the online updating support set. Operator $*$ is the convolution, and $\circ$ is the *Hadamard* product. The scalar $\lambda$ is a learnable regularization parameter.

Since $\{(x, y)\}$ receives various support pair $(x_i, y_i^j)$, we introduced a series of gate function $g$ and $g_i^j$ (dashed line in Figure 3) to control whether we propagate the information and back-propagate the loss of corresponding $\omega_i^j$, $i \in \mathbb{N}, s - M < i \le s, j \in \{p, d, m, s\}$.

$$\mathcal{E}(x_s) * \omega_s = g \left( \sum_{s-M<i<s} g_i^p(\mathcal{E}(x_s) * \omega_i^p), \sum_{\substack{s-M<i\le s \\ j \in \{d,m,s\}}} g_i^j(\mathcal{E}(x_s) * \omega_i^j) \right), \tag{2}$$

where $M$ is the memory size of support set for current section $s$. We implemented $g_i^j$ by linear functions and $g$ by fully-convolutional networks (FCN) in order to learn the importance of each and project the sum to a proper space for following decoder $\mathcal{D}$, respectively.

After that, the differentiable few-shot learner optimizes the target $\arg\min_\omega \mathcal{L}_{\text{shot}}(\omega)$. Following (Bhat et al., 2020), we apply an approximate solution of several iterations of the steepest descent. It can be expressed as $\omega \leftarrow \omega - \alpha^i \beta^i$, where $\beta^i = \nabla\mathcal{L}_{\text{shot}}$ (19) is the gradient and $\alpha^i = \arg\min_{\alpha'} \mathcal{L}_{\text{shot}}(\omega - \alpha'\beta^i)$ (18) is the found step-length on the $i$-th step. In practice, it converges to a satisfactory filter $\omega$ by a handful of $N$ iterations, $i \leq N$.

## 2.4. Training and Inference

For the inference procedure, we follow the pipeline described above. Note that: the online loss $\mathcal{L}_{\text{shot}}$ (1) is being used; the offline representation of $\mathcal{G}, \mathcal{W}(y^{d,m,s})$ could be processed in advance. The resulting segments for each proposal will be agglomerated by post-processing steps (Meirovitch et al., 2019) for the automatic connectome (dashed arrows in Figure 2).

For the training procedure, we simulate the inference procedure for our end-to-end architecture. To ensure robustness, we randomly sample $S$ serial sections as each sequence from the whole tissue. All network parameters $\theta$ (except $\mathcal{B}_\theta$) are trained by the per-sequence loss $\mathcal{L}_{\text{film}}$ (3). Our box extractor $\mathcal{B}_\theta$ is trained by freezing other parameters in the network and minimizing the segment loss (Berman et al., 2018) $\mathcal{L}_{\text{seg}}\left(\mathcal{D}\left(\mathcal{E}(x_0), \mathcal{B}_\theta(\mathcal{E}(x_0), y_0^b)\right), y_0^*\right)$.

$$
\begin{aligned}
\mathcal{L}_{\text{film}}(\theta) &= \frac{1}{S-1} \sum_{s=1}^{S-1} \mathcal{L}_{\text{seg}}\left(\mathcal{D}_\theta\left(\mathcal{E}_\theta(x_s), \mathcal{E}_\theta(x_s) * \omega_{s-1}\right), y_s^*\right), \text{ where} \\
\omega_0 &= \mathcal{F}_{\mathcal{B},\mathcal{G}_\theta,\mathcal{W}_\theta}\left(\left\{(x_0, y_0^j)\right\}_{j \in \{b,d,m,s\}}\right) \text{ or } \omega_s = \mathcal{F}_{\mathcal{G}_\theta,\mathcal{W}_\theta}\left(\left\{(x_i, y_i^j)\right\}_{\substack{s-M<i\leq s \\ j \in \{p,d,m,s\}}}\right),
\end{aligned}
\tag{3}
$$

where $y^*$ denotes the ground truth, and $\mathcal{G}, \mathcal{W}$ could be learned through the differentiable $\mathcal{F}$. As mentioned in (2), $i$ traverses sections and $j$ traverses types in the support set.

## 3. Experiments

We evaluate our method on multiple connectomic datasets across various animals and different scales. These open datasets (Table 1) are listed following: **SNEMI** (Arganda-Carreras et al., 2015), **Kasthuri** (Kasthuri et al., 2015), **J0126** (Januszewski et al., 2018), and **Joesch** (Joesch et al., 2016). Note that: Kasthuri is the superset of full SNEMI dataset; J0126 is kindly provided by (Sheridan et al., 2022); Joesch is in the region of *Starburst Amacrine* cell (SAC) plexus selected from $\sim 4.5 \times 10^5 \mu m^3$, stained with *APEX2* tags.

The main challenges of 3D reconstruction of connectomic datasets could be briefly described as the large scale of data, the huge variance of different areas (Figure 4C), and other issues introducing the merged or split errors. We employ the VI metric to distinguish the integrity of segmentation, the ERL-based metrics (more details in Appendix B) to check tracking errors, and the FLOPs to examine the feasibility in practice.

We use the widely-used *PyTorch Connectomics* (Lin et al., 2021) repository with default configurations from the official document, which could be easy to reproduce related

Table 1: Comparison on datasets used for experiments.

| Dataset (abbreviated) | Resolution ($x, y, z$-axis) | Training Data (approximate size) | Testing Data (approximate size) |
|---|---|---|---|
| SNEMI | $6, 6, 30\ nm$ | 1/2 dense seg. $\sim 50\mu m^3$ | the remaining 1/2 |
| Kasthuri | $6, 6, 30\ nm$ | the whole SNEMI | 1 sparse seg. $\sim 10^6 \mu m^3$ |
| J0126 | $9, 9, 20\ nm$ | 33 dense seg. $\sim 200\mu m^3$ | 50 skel. $\sim 97mm, 10^6\mu m^3$ |
| Joesch | $4, 4, 30\ nm$ | 1/2 sparse seg. $\sim 3500\mu m^3$ | the remaining 1/2 |

experiments on the ultrastructure segmentation, long-range affinities and watershed. For experiments on Kasthuri, we primitively borrow the network parameters trained on SNEMI and fine-tune them. Training and inference codes will be released upon publication.

### 3.1. Ablation Studies

Here, we analyze the impacts of the various awareness introduced into our proposed II-CATS architecture. The ablation studies are performed on the hold-out testing set, *i.e.*, the remaining 1/2 densely labeled 6x3x3 $\mu m^3$ stack in SNEMI. For simplicity, we use the same pretrained backbone ResNet-50 (He et al., 2017) for our models in Table 2.

**Baseline** model FFN-a/b constitutes the open source version *Flood-Filling Networks* (Januszewski et al., 2018) (FFN), where the FCN blocks could be simply plugged in the color encoding of transfer mechanism from (Meirovitch et al., 2019) (FFN-3C), or the mitochondrion and synapse information (FFN w/ *mito.*, *syn.*). To achieve this, we put parallel input channels for these subcellular *pixel*-level candidates, increase the number of residual modules, and dilate the field of view (FoV) size in the original FFN.

**TS** model directly explores the *quasi-dense* matching proposal of neurons. Our few-shot learner $\mathcal{F}$ only receives the output of the tracker $\mathcal{T}$, *i.e.*, the initial support set is $\{(x, y^b)\}$.

**CATS** model further exploits the cellular description $y^d$ (long-range affinities, shape direction vectors and shape sizes). We also use the differentiable $\mathcal{F}$ to train the underlying representation for target filter $\omega$ in an end-to-end manner.

**II-CATS** models are verified separately (w/ *mito.*, *syn.*) and simultaneously. The information on interposition mitochondrion provides a substantial decrease in VI, which is similar to previous reports on FFN. Besides, the information of synaptic cleft partition contributes to *merged* errors, extending the ERL by additional $(x, y^{m,s})$ support pairs.

The results of the bERL metric (Appendix B.2) emphasize the need for connectivity robustness around biological structures. It suggests that we could add explicit modeling of bERL concept to the training procedure of $\mathcal{F}$ (w/ *mito.*) in future work.

### 3.2. Comparison

In this subsection, we extend our comparison from the small-scale SNEMI to rather large-scale datasets, which could be more meaningful in practice, shown in Table 1, 3. As for the decreased VI on the Kasthuri dataset, II-CATS successfully obtains the predicted subcellular

Table 2: Ablative analysis of cellular, intra- and inter-cellular awareness modules on SNEMI dataset. The unit of all kinds of ERL-based metrics is $nm$. The top-2 ranking places are in bold.

| Method | VI ↓ | ERL ↑ | bERL ↑ |
|---|---|---|---|
| FFN-a | 0.938 | 2771 | 3313 |
| FFN-b | 0.953 | 2816 | 3305 |
| FFN-3C | 1.152 | **2925** | 3464 |
| FFN w/ *mito.* | 0.920 | 2905 | 3472 |
| FFN w/ *syn.* | 0.991 | 2766 | 3249 |
| TS | 1.003 | 2088 | 2191 |
| CATS | 0.917 | 2267 | 2506 |
| CATS w/ *mito.* | **0.761** | 2332 | **3489** |
| CATS w/ *syn.* | 0.890 | 2827 | 3025 |
| II-CATS | **0.709** | **3056** | **3690** |

Table 3: Comparison on the Kasthuri, J0126, and the stained Joesch dataset.

| Method | Kasthuri | | J0126 | | Joesch | | |
|---|---|---|---|---|---|---|---|
| | VI ↓ | tERL ↑ | tERL ↑ | nERL ↑ | VI ↓ | tERL ↑ | FLOPs ↓ |
| Watershed | | | | | 1.505 | 3264 | $\sim 10^{17}$ |
| FFN-a | 1.898 | **8964** | **16886** | 8209 | 0.922 | 5931 | $> 10^{19}$ |
| CATS | 1.835 | 8561 | 14942 | **9361** | **0.820** | **5956** | $\sim 10^{18}$ |
| II-CATS | **1.725** | 8819 | | | | | |

results, which were only trained on a small-scale SNEMI dataset. As for the two ERL results on J0126, CATS also reaches comparable results to the original FFN method.

The results of the nERL metric (Appendix B.3) considered the scale factor among the different parts of skeletons, which needs to be closely followed during whole-cell proof-reading. The visualization on a part of the J0126 dataset (neuron $\mathbf{N_3}$ in Appendix A) demonstrates that our method almost gets rid of small fragments and tunnels the FFN results still exists, even though both of methods have masked out the nucleus region.

In particular, we also do our experiments on the target cells that possessed typical staining characteristics, illustrated in Table 3. Due to the interference between the darkened areas and original membrane boundaries, methods tracking from proposals (Figure 1b) usually excel the other methods segmenting from membrane descriptors (Figure 1a). However, our proposed method outperforms both of them on these cells of the Joesch dataset. We also estimate qualitative computational complexity analysis, which infers that our method (CATS) is at least 10 times faster compared to FFN-a.

Further details are provided in Appendix D. Note that: the nERL metric needs somas or nuclei, which is unavailable for the original SNEMI stack (Table 2). We also do not have the ground truth of mitochondria and synapses in Kasthuri, J0126 and Joesch datasets, so

we can not test the bERL metric in these datasets right now. Next step, we are looking forward to building our own dataset, which will contain everything we need.

## 4. Discussion

We present II-CATS, a series of connectomic few-shot learning methods based on our proposed novel paradigm. First and foremost, this new way of thinking guides us to exploit the cellular, intra- and inter-cellular priori for more robust neuron tracking and segmentation results. After that flash of insight, we implement a multiple object tracker with *quasi-dense* anchors and a single object decoder with *learning to learn* parameters in order to achieve that paradigm in practice. Furthermore, we use the VI and ERL metrics along with the biologically implied bERL and nERL metrics to evaluate our methods.

Among open source volume electron microscopy (vEM) datasets[2] for connectomics, our section-by-section tracking and segmentation method works well on anisotropy datasets (Table 1, serial-section vEM methods), by contrast, we could directly fly through 3D isotropy datasets (block-face vEM methods) as an extension (Schmidt et al., 2022) in future works. Besides, it may also be essential to prune the complex pipeline and give more protocols in various datasets for reproducibility. Moreover, we found bERL (w/ *mito.*) is exceptionally useful, so that the differential version of bERL may contribute to the skeleton-level tracking. Finally, resent works (Bae et al., 2021; Turner et al., 2022) has revealed the necessity of functional connectomics, which needs to be analyzed in conjunction with fluorescence imaging[3] and behavioral recordings[4] (Chen et al., 2021) to obtain more meaningful neuron circuit modeling data in brain projects (Ngai, 2022).

To sum up, our methods have the advantages on small, large and stained vEM datasets, meanwhile having the potential to generalize to more complex 3D reconstruction.

## Acknowledgments

This work was supported by the National Natural Science Foundation of China (32171461), the STI 2030-Major Projects (2021ZD0204500 and 2021ZD0204503), and the Strategic Priority Research Program of Chinese Academy of Science (XDA16021104). We also thank the Transdisciplinary Platform of Brain Functional Connectome and Brain-inspired Intelligence in Huairou Science City in Beijing for providing technical support and device resources.

---

2. https://github.com/JackieZhai/awesome-vem-datasets

3. https://github.com/JackieZhai/awesome-fm-datasets

4. https://github.com/JackieZhai/awesome-behavior-datasets

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

# Appendix A. Qualitative Results

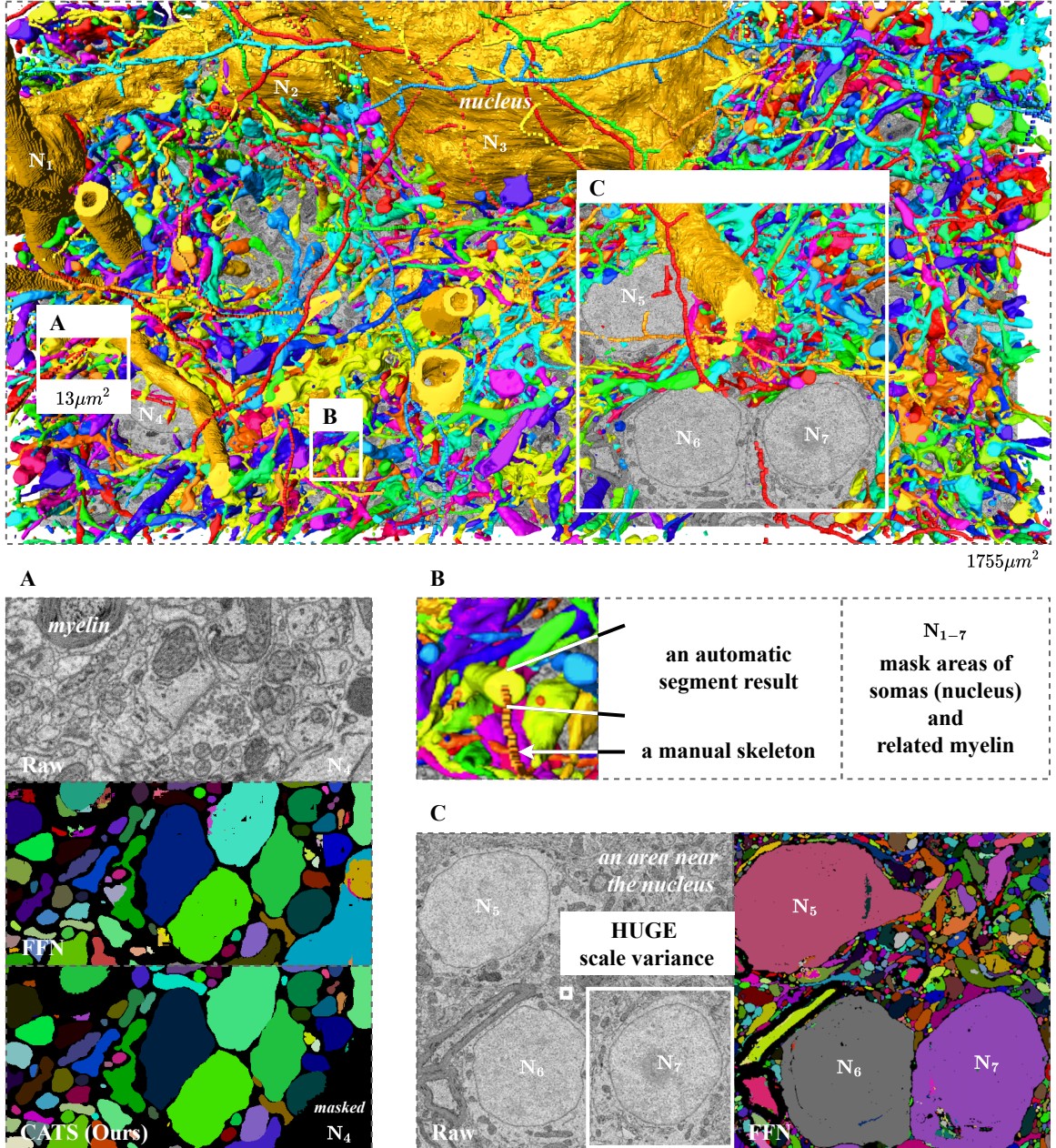

Figure 4: Visualizations of a part of dataset J0126. **A:** Qualitative comparison between FFN and CATS (Ours) from an about $13\mu m^2$ patch of $1755\mu m^2$. **B:** We also render $4\mu m$ automatic prediction of segments as well as $14\mu m$ somas and myelin of $\mathbf{N_{1-3}}$, and manual skeletons upon the section, demonstrating the results of our results. **C:** Nearby the nucleus, other tracking method generates split errors due to the huge scale variance, which could be measured by our nERL metric.

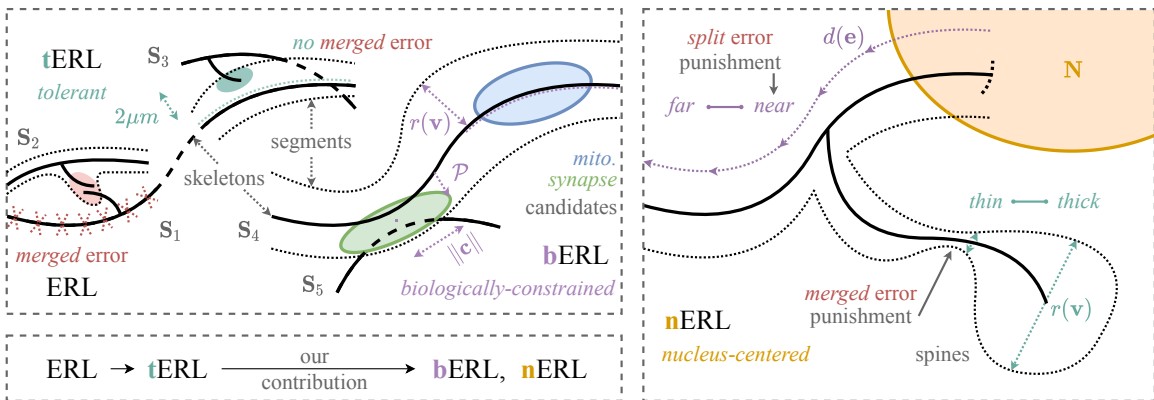

Figure 5: Sketches of the ERL, tERL, bERL and nERL metric. These ERL-based metrics are evaluated by proofread skeletons (solid line) and predicted segments (dotted envelope line). **Left:** Our proposed bERL punishes errors nearby mitochondria (blue) and synaptic clefts (green). **Right:** The soma or nucleus (yellow) and its neurites indicate the different error punishment of nERL from near to far and from thin to thick.

## Appendix B. Metrics

Metrics in connectomics are essential for the error evaluation of automated reconstruction. This condition is mainly because of the catastrophic man-hour consumption of proofreading (Scheffer et al., 2020). Intuitively, the classic and adapted random error (ARAND (Lee et al., 2017)) and variation of information (VI (Meilă, 2003)) was introduced to measure the *voxel*-level error between segments. Then, it is reasonable for tolerant edit distance (TED (Funke et al., 2017)) and other topological errors (Hu et al., 2020) to focus on *shape*-level uniformity, which reduces the side effect of a tiny distinction. To assess the connectivity of reconstructed neural networks, synaptic *cleft*-level partner identification could be examined by neural reconstruction integrity (NRI (Reilly et al., 2018)). According to the reported false merges and false splits in the process of proofreading, the inter-error distance (IED (Berning et al., 2015)), expected running length (ERL (Januszewski et al., 2018; Wei et al., 2021)) and min-cut metric (MCM (Sheridan et al., 2022)) utilize *skeleton*-level tracings to determine the error rate.

**The evolution of large-scale connectomic metrics matters.** It may be risky that only extrapolate from the benchmark evaluation of limited data, instead, the large-scale datasets are robust to estimate real-world accuracy of the neural circuit reconstruction (Lee et al., 2017). The original ERL-based metrics should have been suitable for large-scale because they only need manual skeletons (instead of segments) for sparse reconstruction (Liu et al., 2021), but they are missing key biological constraints (*e.g.*, the nuclei information only occurs in large-scale volumes).

**The insight into subcellular structures matters.** Nuclei are located in the center of neurons (Jiang et al., 2019). Synapses (and their clefts) extract the inter-neuron connections (Parag et al., 2018; Liu et al., 2022). Mitochondria (and their unique texture) reflect the intra-neuron stability (Xiao et al., 2018; Hong et al., 2022, 2023). Other structures such as myelin and glia may also helpful (Dorkenwald et al., 2017). From these perspectives, we think the integrated evaluation of neurons ERL needs to embrace a priori of organelles

and a structure centered on nuclei. Also, some recent experiments showed similar ERL results (Schmidt et al., 2022). Thus, we extended the original ERL metric by focusing on biological constraints (bERL) and spanning from the nucleus to spines (nERL).

Considering the proofread or generated skeletons $\mathbf{S} \in \mathbb{S}$ (ground truth) and predicted segments $\mathbf{L} \in \mathbb{L}$ (labeled supervoxel), we define the mapping function $m(\mathbf{e})$ for any edge $\mathbf{e} = \{\mathbf{v}, \mathbf{v}'\} \in \mathbf{S}$, and the length function $f(\mathbf{S}, \mathbf{L})$ for the *correct* paths at the intersection of $\mathbf{S}$ and $\mathbf{L}$. Give:

$$m(\mathbf{e}) = \begin{cases} 0, & \mathbf{v} \in \mathbf{L_1}, \mathbf{v}' \in \mathbf{L_2} \\ \|\mathbf{e}\|, & \mathbf{v}, \mathbf{v}' \in \mathbf{L} \end{cases}, \tag{4}$$

$$f(\mathbf{S}, \mathbf{L}) = \begin{cases} 0, \exists \mathbf{v_1}, \mathbf{v_2} \in \mathbf{L}, \mathbf{v_1} \in \mathbf{e_1} \in \mathbf{S_1}, \mathbf{v_2} \in \mathbf{e_2} \in \mathbf{S_2} \\ \sum_{\mathbf{e} \in \mathbf{S}, \mathbf{v} \in \mathbf{L}} m(\mathbf{e}), \quad \text{otherwise} \end{cases} \tag{5}$$

where each skeleton $\mathbf{S}$ is defined as a set of edges, each segment $\mathbf{L}$ is defined as a set of vertices, and $\|\mathbf{e}\|$ denotes the length of edge. The $m(\mathbf{e}) = 0$ shows the *split* paths, and the $f(\mathbf{S}, \mathbf{L}) = 0$ shows the *merged* paths. The original ERL (Januszewski et al., 2018) metrics are:

$$\mathrm{ERL}(\mathbf{S}) = \frac{1}{\|\mathbf{S}\|} \sum_{\mathbf{L} \in \mathbb{L}} f(\mathbf{S}, \mathbf{L})^2, \tag{6}$$

$$\mathrm{ERL}(\mathbb{S}) = \frac{1}{\sum_{\mathbf{S} \in \mathbb{S}} \|\mathbf{S}\|} \sum_{\mathbf{S} \in \mathbb{S}} \sum_{\mathbf{L} \in \mathbb{L}} f(\mathbf{S}, \mathbf{L})^2, \tag{7}$$

where $\|\mathbf{S}\| = \sum_{\mathbf{e} \in \mathbf{S}} \|\mathbf{e}\|$ denotes the length of skeleton.

### B.1. tERL

However, the original ERL metric has a few drawbacks. For example, it emphasizes merge errors disproportionally (Sheridan et al., 2022), and may assign unexpected zero run length from skeletonization artifacts. Wei *et al.* (Wei et al., 2021), therefore, designed a tolerance threshold $\tau$ of *merged* vertices (around 2 $\mu m$ in length) for the more robust tERL, which could substitute $f_{\mathrm{t}}(\mathbf{S}, \mathbf{L})$ for $f(\mathbf{S}, \mathbf{L})$ in the original ERL:

$$f_{\mathrm{t}}(\mathbf{S}, \mathbf{L}) = \begin{cases} 0, & |\mathbf{T}| > \tau \\ \sum_{\mathbf{e} \in \mathbf{S}, \mathbf{v} \in \mathbf{L}} m(\mathbf{e}), & \text{otherwise} \end{cases}, \tag{8}$$

where $\mathbf{T} = \{(\mathbf{v_1}, \mathbf{v_2}) \mid \mathbf{v_1}, \mathbf{v_2} \in \mathbf{L}, \mathbf{v_1} \in \mathbf{e_1} \in \mathbf{S_1}, \mathbf{v_2} \in \mathbf{e_2} \in \mathbf{S_2}\}$, which is the vertex pair of *merged* paths.

The following two subsections show some of our efforts, which have been dedicated to extending the original ERL metric to the biological implications (Figure 5).

### B.2. bERL

We design the biologically-constrained expected run length (bERL), which could substitute $m_{\mathrm{b}}(\mathbf{e})$ and $f_{\mathrm{b}}(\mathbf{S}, \mathbf{L})$:

$$m_{\mathrm{b}}(\mathbf{e}) = \begin{cases} -\epsilon, & \mathbf{v} \in \mathbf{V}_{mito}, \mathbf{v} \in \mathbf{L_1}, \mathbf{v}' \in \mathbf{L_2} \\ \|\mathbf{e}\|, & \mathbf{v}, \mathbf{v}' \in \mathbf{L} \\ 0, & \text{otherwise} \end{cases}, \tag{9}$$

$$f_{\mathrm{b}}(\mathbf{S}, \mathbf{L}) = \begin{cases} 0, & |\mathbf{T}| > \tau \quad \text{or} \quad \exists \mathbf{v} \in \mathbf{T}, \mathbf{v} \in \mathbf{V}_{syna} \\ \sum_{\mathbf{e} \in \mathbf{S}, \mathbf{v} \in \mathbf{L}} m_{\mathrm{b}}(\mathbf{e}), & \text{otherwise} \end{cases}, \tag{10}$$

where the constraints of $\mathbf{V}_{mito}$ and $\mathbf{V}_{syna}$ are shown as:

$$\forall \mathbf{v} \in \mathbf{V}_{mito}, \exists \mathbf{u} \in \mathbf{U}_{mito}, \|\mathbf{e}\| := \|\mathbf{v} - \mathbf{v}'\|$$
$$\text{s.t.} \quad \|\mathcal{P}_{\mathbf{e}}(\mathbf{u}) - \mathbf{v}\| \leq \|\mathbf{e}\| + r(\mathbf{v}) \tag{11}$$
$$\|\mathcal{P}_{\mathbf{e}}(\mathbf{u}) - \mathbf{u}\| \leq r(\mathbf{v})$$

$$\forall \mathbf{v} \in \mathbf{V}_{syna}, \exists \mathbf{u} \in \mathbf{U}_{syna}, \|\mathbf{c}\| := \max_{\mathbf{u}} \left\| \mathbf{u} - \frac{\sum \mathbf{u}}{|\mathbf{U}_{syna}|} \right\|$$
$$\text{s.t.} \quad \|\mathcal{P}_{\mathbf{c}}(\mathbf{v}) - \mathbf{v}\| \leq \|\mathbf{c}\| + r(\mathbf{v}) \tag{12}$$
$$\|\mathcal{P}_{\mathbf{c}}(\mathbf{v}) - \mathbf{u}\| \leq r(\mathbf{v})$$

where $\mathbf{U}_{mito}$ and $\mathbf{U}_{syna}$ are existing mitochondrion and synapse candidates. Operator $\mathcal{P}_{\mathbf{i}}$ is the projection into $\mathbf{i}$. We define the radius function $r(\mathbf{v})$ to describe the average radius of vertices and $\|\mathbf{c}\|$ to describe the radius of fitted synaptic clefts. The fitted cleft could be abstracted to a center point and a surface with a radius.

The bERL metric could be quickly deployed when one has the proofread mitochondrion or synaptic cleft instances, which are far easier to acquire than all of the neuron instances. Moreover, this metric screens subcellular related fragments and checks them out, which will be helpful for optimizing the areas consisting of biological information.

### B.3. nERL

We further design the nucleus-centered expected run length (nERL), which could substitute $m_{\mathrm{n}}(\mathbf{e})$ and $f_{\mathrm{n}}(\mathbf{S}, \mathbf{L})$. We define the logarithmic distance function $d(\mathbf{e})$ to measure the crow flight between $\mathbf{e} \in \mathbf{S}$ and the precomputed nucleus center of $\mathbf{S}$. The $r(\mathbf{v})$ and $d(\mathbf{e})$ share the same unit.

$$m_{\mathrm{n}}(\mathbf{e}) = \begin{cases} 0, & \mathbf{v} \in \mathbf{L_1}, \mathbf{v}' \in \mathbf{L_2} \\ r(\mathbf{v}) \cdot \|\mathbf{e}\|, & \mathbf{v}, \mathbf{v}' \in \mathbf{L} \end{cases}, \tag{13}$$

$$f_{\mathrm{n}}(\mathbf{S}, \mathbf{L}) = \begin{cases} 0, & |\mathbf{T}| > \tau \\ \sum_{\mathbf{e} \in \mathbf{S}, \mathbf{v} \in \mathbf{L}} \frac{m_{\mathrm{n}}(\mathbf{e})}{d(\mathbf{e})}, & \text{otherwise} \end{cases}, \tag{14}$$

Besides Figure 5 the sketches of our metrics, we also give real examples from the SNEMI dataset (same as Figure 2). The extra figures below are viewed in the *Neuroglancer*[5].

---

5. https://github.com/google/neuroglancer and https://neuroglancer-demo.appspot.com

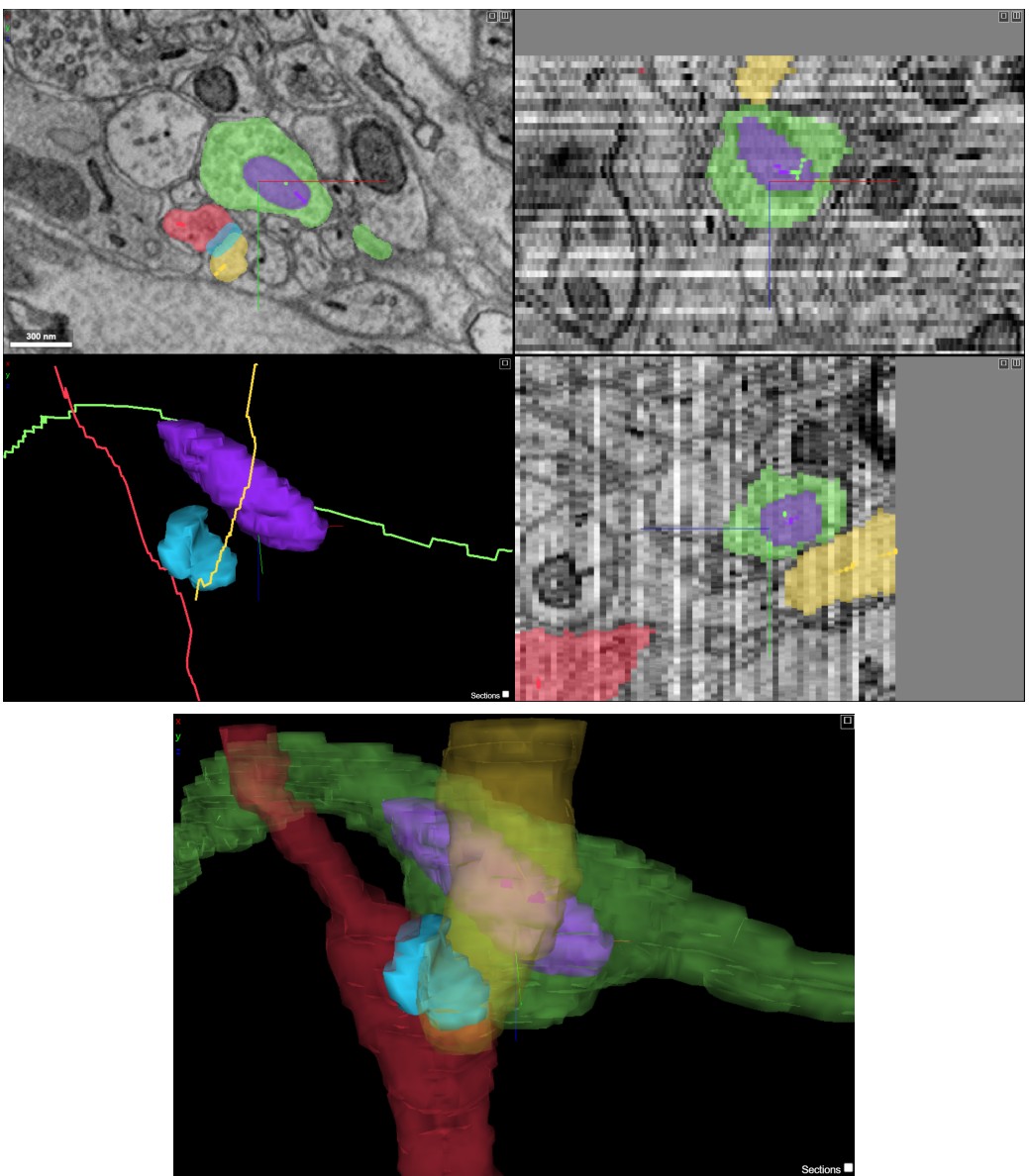

Figure 6: **Top:** Three views and 3D rendering of the relevant area in Figure 2 bottom-right. Three lines indicate the skeleton of neurites. Purple represents a mitochondrion, and blue represents a synaptic cleft. **Bottom:** 3D rendering of the relevant area in Figure 2 bottom-right. Purple represents a mitochondrion, and blue represents a synaptic cleft. Neurites are translucent.

The two Figure 7 and 8 illustrates one of the real situation of bERL in Figure 5.

In addition, we would like to describe the equations of our proposed metrics step by step in detail.

In Equation (9), based on the original $m(\mathbf{e})$, a more severe penalty is given to the vertex $\mathbf{v}$ in $\mathbf{V}_{mito}$. The penalty term $\epsilon$ can be given according to the reliability of the mitochondrial segmentation results in the dataset.

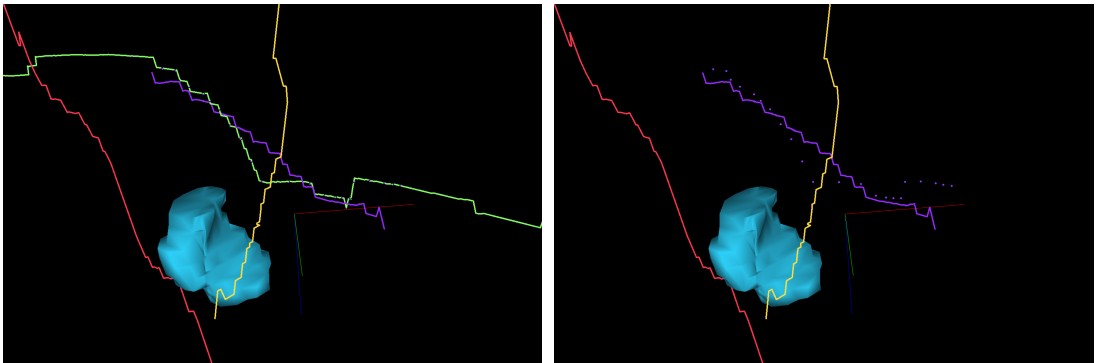

Figure 7: The purple line shows the skeleton of this mitochondrion, and the points of them will be projected to corresponding neurite ($\mathcal{P}_\mathbf{e}(\mathbf{u})$ in Equation (11)), see the purple points).

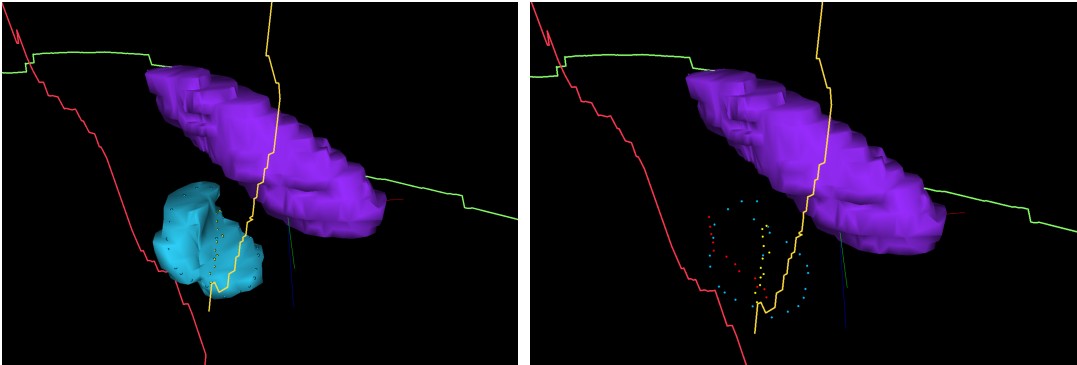

Figure 8: The blue points show the counter and the center point of this synaptic cleft, which are the results of fitting processing ($\mathbf{c}$ in Equation (12)). One neurite (yellow line) skeleton projects to the synaptic cleft ($\mathcal{P}_\mathbf{c}(\mathbf{v})$ in Equation (12), see the yellow points), another neurite (red line) skeleton also projects to the synaptic cleft (see the red points), and they construct a connection of neurites in the connectome.

In Equation (10), except for the case where the penalty is greater than the tolerance threshold $\tau$, the vertex $\mathbf{v}$ in $\mathbf{V}_{syna}$ is directly penalized. Such drastic measures can significantly reflect error conditions around the synapse.

Equation (11) and Equation (12) present a correspondence, and both show how we use the projection operator to achieve the above idea. It should be noted that we abstract the synaptic cleft as a kind of disk by fitting and saving it as three parameters of position $\mathbf{c}$, direction $\bar{\mathbf{c}}$, and radius $\|\mathbf{c}\|$, which is convenient for the subsequent projection process.

The significance of bERL is that for the areas that biologists pay more attention to, such as the location of these important subcellular structures, we need to extract relevant sites through automated algorithms and further optimize them.

In Equation (13), we achieve a strategy of increasing the *merged* error penalty for thin places (like spines) by introducing the radius $r(\mathbf{v})$ of the skeleton.

In Equation (14), we increase the *split* error penalty for objects that are too close by introducing $d(\mathbf{e})$. It is important to note that the reason this function varies logarithmically

is that we do not want to cause objects that are too far away to completely lose their contribution to the ERL.

The importance of nERL is that it gives us a perspective to evaluate the tracking IoU of different FoV locations of skeletons on a larger scale, which cannot be reflected in previous small-scale datasets.

## Appendix C. Architecture Details

### C.1. Quasi-Dense Tracker ($\mathcal{T}$)

As shown in (Pang et al., 2021), given the key section for training, we randomly select a reference section from the neighborhood. Assume the embedding of an object instance on the key section is denoted as $\mathbf{k}$. An instance on the reference section is defined as positive $\mathbf{r}^+$ if two anchors with IoU higher than 0.7 are associated with the same instance, and negative $\mathbf{r}^-$ otherwise. We can use the non-parametric softmax with cross-entropy to optimize the embedding of the object $\mathbf{k}$ by $\mathcal{L}_{\mathrm{match}}$:

$$\mathcal{L}_{\mathrm{match}} = \log \left( 1 + \sum_{\mathbf{r}^+} \sum_{\mathbf{r}^-} \exp \left[ \mathbf{k} \circ \mathbf{r}^- - \mathbf{k} \circ \mathbf{r}^+ \right] \right), \tag{15}$$

$$\mathcal{L}_{\mathrm{cosine}} = \left( \frac{\mathbf{k} \circ \mathbf{r}^+}{\|\mathbf{k}\| \circ \|\mathbf{r}^+\|} \right)^2 + 3 \left( \frac{\mathbf{k} \circ \mathbf{r}^-}{\|\mathbf{k}\| \circ \|\mathbf{r}^-\|} \right)^2, \tag{16}$$

$$\mathcal{L}_{\mathrm{embed}} = \gamma_3 \mathcal{L}_{\mathrm{match}} + \gamma_4 \mathcal{L}_{\mathrm{cosine}}, \tag{17}$$

where loss weights are set to $\gamma_1 = 1.0, \gamma_2 = 1.0, \gamma_3 = 0.25, \gamma_4 = 1.0$ by default. Note the loss $\mathcal{L}_{\mathrm{cosine}}$ aims to constrain the logit magnitude and cosine similarity instead of improving the performance. We select 128 anchors from the key frame as training samples and 256 anchors from the reference frame with a positive-negative ratio of 1.0 as contrastive targets. We use 4 convolution and 1 fully-connection head with group normalization to extract feature embeddings. The channel number of embedding features is set to 256 by default. We train our models with a total batch size of 16 and an initial learning rate of 0.02 for 20 epochs, decreasing the learning rate by 0.1 after every 10 epochs.

### C.2. Internal Representation Extractor ($\mathcal{B}, \mathcal{G}, \mathcal{W}$)

By the way, we noticed that local shape descriptors (LSD) also tried to expand their channels (Sheridan et al., 2022) for encoding (10-channel descriptors, not limited to 1-channel boundaries or 3-channel affinities), which may share similar concepts with our internal representation extractor $\mathcal{B}, \mathcal{G}, \mathcal{W}$ (32-channel representation).

### C.3. Few-shot Learner ($\mathcal{F}$)

For simplicity, we redefine the corner of $\omega$ here as the iterations of the steepest descent step. The differentiable few-shot learner optimizes the target $\omega = \mathcal{F}(\{(x, y)\})$, which is $\arg\min_{\omega'} \mathcal{L}_{\mathrm{shot}}(\omega')$. The well-known closed-form solution of this problem requires extensive matrix multiplications. Following (Bhat et al., 2020), we apply an approximate solution of several iterations of steepest descent, which can be expressed as $\omega^{i+1} = \omega^i - \alpha^i \beta^i$ and:

$$\alpha^i = \frac{\|\beta^i\|^2}{\sum_t \|\mathcal{W}(y) \circ (x * \beta^i)\|^2 + \lambda\|\beta^i\|^2}, \tag{18}$$

$$\beta^i = \sum_t x *^T \left(\mathcal{W}^2(y) \circ (x * \omega^i - \mathcal{G}(y))\right) + \lambda\omega^i, \tag{19}$$

where $\beta^i = \nabla\mathcal{L}_{\text{shot}}$ is the gradient and $\alpha^i = \arg\min_{\alpha'} \mathcal{L}_{\text{shot}}(\omega - \alpha'\beta^i)$ is the found step-length on the $i$-th step. Operator $*^T$ is the transposed convolution.

The implemented few-shot learner module $\mathcal{F}(\omega^0) = \omega^N$ predict satisfactory filter $\omega$ by a handful of $N$ iterations. In practice, we would try various meta-learning-based steepest descent methods to decide which one to use, including Newton's method when minimizing the KL-divergence loss (Battiti, 1992).

### C.4. Segmentation Decoder ($\mathcal{D}$)

We take the mirrored 3D residual networks as the segmentation decoder, which is similar to U-Net. The decoder receives the output of the target $\omega$ convolution along with features from encoder $\mathcal{E}$ to predict accurate segmentation masks. We used pretrained ResNet-50 to balance simplicity and accuracy. The encoder could be divided as four residual blocks $\mathcal{E}^{1,2,3,4}$, so that the corresponding output feeds to $\mathcal{D}^{1,2,3,4}$.

For each decoder block $\mathcal{D}^i$, we designed three parts for feature projection and fusion. First, the encoding features need to project to a lower-dimensional representation, which could be concatenated with the $\omega$ representation. Second, the concatenated features are processed by convolutional layers followed by a small residual block. Third, the resulting features are then merged with features from deeper decoder $\mathcal{D}^{i+1}$ by a channel attention block. Note that the interpolation ratio of $\omega$ representation for each $\mathcal{D}^i$ is equal to $2^{3-i}$.

### C.5. Training and Inference

For training all networks except $\mathcal{B}$: We use the sequence length $S = 7$ sections; The number of steepest descent iterations $N = 5$ for the initial segment and $N = 2$ for subsequent segments; We use Adam to train; We trained for first 100k iterations with backbone fixed (Mask R-CNN pretrained ResNet-50) and another 100k iterations. For training box extractor $\mathcal{B}$: We fixed all other networks and only trained for 50k iterations.

For the memory of the support set during the inference procedure, we reduce the impact of the previous section support pairs by a decay function with parameters $\eta$. Note that the initial supports have a low impact by default because boxes are weaker supervisor than masks.

## Appendix D. Implementation Details

We try not to use any meaningless tricks for network architecture design. Some of the subsequent network settings are based on previous research (Januszewski et al., 2018; Bhat et al., 2020; Pang et al., 2021).

### D.1. FFN-a/b

We used the open source on the official repository[6] in 2020 as the FFN-a/b method without agglomeration steps. The results of this code appear in Table 2, 3.

For the J0126 dataset, we use FFN FoV size 33x33x17, FoV step size 8x8x4, 9 residual modules, initial FoV fill value 0.05, initial FoV seed value 0.95, FoV movement threshold 0.9, image irregularity detection (tissue classification, patch-wise cross-correlation), consensus input segmentations of 36x36x40, 18x18x20, 18x18x20, 9x9x20, 9x9x20.

For the SNEMI dataset, we use CLAHE and mirror padding for data pre-processing, FFN FoV size 33x33x17, FoV step size 8x8x4, 9 of residual modules, initial FoV fill value 0.5, initial FoV seed value 0.95, FoV movement threshold 0.6, consensus input segmentations of 6x6x30. And the superset Kasthuri[7] uses the same configuration.

Note that the qualitative result of Figure 4 A is a patch in FFN results from Funke *et al.*, which are kindly provided by (Sheridan et al., 2022).

### D.2. FFN-3C

For the so-called color encoding mechanism we use for FFN, it is named cross-classification clustering (Meirovitch et al., 2019). Their goal is to extend a single-object classification from one image to the next so as to simultaneously classify pixels for an a priori unknown set of object labels.

### D.3. FFN w/ *mito.*, *syn.*

For add another channel for *mito.* or *syn.* on SNEMI dataset, we also use CLAHE and mirror padding for raw images along with the voxel-level candidate segmentations. The FFN then has the FoV size 65x65x33, FoV step size 8x8x4, 12 of residual modules, initial FoV fill value 0.5, initial FoV seed value 0.95, FoV movement threshold 0.6, consensus input segmentations of 6x6x30.

### D.4. U-Nets

We not only use the long-range affinities, but also implement some of the LSD (Sheridan et al., 2022). To be specific, we find the layer 0 to 2 and 9 are special for our tasks in their code[8] of generating 3D neuron LSD.

The layer 0 to 2 denote the Gaussian point offset to shape center and layer 9 denotes the shape size. The LSD is $lsd^y(v) = (\hat{s}(v), \hat{m}(v) - v, \hat{c}(v))$, and we use the first 2 components to formulate an auxiliary learning task that complements the prediction of affinities.

We use difference settings such as SNEMI-Affinity-ResNet, SNEMI- Affinity-UNet, SNEMI-Affinity-UNet-LR, SNEMI-Affinity-UNet-MER, Zebrafinch-Affinity-UNet, Zebrafinch-Affinity-UNet-MER, Distance-Transform-Quantized, and Multi-class-Semantic-Seg. These are freely provided by Lin *et al.* on the documents of *Pytorch Connectomics*[9].

---

6. https://github.com/google/ffn/tree/0570a55d75cae3a1ef1bedd5fb98a28f4dc68ef1

7. https://lichtman.rc.fas.harvard.edu/vast

8. https://github.com/funkelab/lsd

9. https://github.com/zudi-lin/pytorch_connectomics

