# OpenReview forum: "Intra- and Inter-Cellular Awareness for 3D Neuron Tracking and Segmentation in Large-Scale Connectomics"
_MIDL.io/2023/Conference — MIDL 2023 Poster_

### Official Review · Reviewer_5phi · 2023-02-02

**Confidence:** 3
**Preliminary Rating:** 4
**Recommendation:** Poster

**Summary:**

Authors are suggesting a method for tracking and segmenting neurons in connectomics.  A few shot learner utilizes a set of different pieces of information obtained from proposal generator, cellular descriptors, mitocondria candidates and synapse candidate. using all of these, they perform neuron segmentation and tracking

**Strengths:**

- Authors' method seems to out perform the state of the art on several datasets.
- They performed a wide study on the literature and provide a rich list of references
- Putting together all the modules seems to be a challenging task which authors have realized in this work

**Weaknesses:**

- Methodology does not seem to describe the whole system. Authors do not provide an overview of the interactions between different components properly
- Information is missing in evaluation section regarding the training and data split
- Figures can improve a lot
- Reading the paper is not easy
- Losses are not completely defined




**Deanonymize Review:**

no

**Detailed Comments:**

The paper is suggesting a new methodology (if I get it correctly) however most of main body of the text is spent on problem statement, state of the art, and evaluation metrics (TED, ERL, bERL, MCM, VI, IED, nERL, list continues with 4 more pages in appendix). Some parts might have a lower priority and the reader can refer to references or the appendix for further details. Authors could have shortened the introduction and maybe excluded Figure 1 to have more space for the methodology. (3 pages of intro vs. ~1 page of methods)

In section "2.1 Overall Architecture", Authors introduce 6 different modules in their system but only describe the few shot learner and everything else is moved to appendix. However, it is absolutely necessary to address all components of the system and how they interact with each other in main section of the paper and not in the appendix. Even if it is done with a few sentences.

It is unclear how did Authors evaluate their method. Not in terms of metrics but addressing how was the test set selected? Was it hold out or cross validated? In both cases probably standard deviations need to be reported in the result tables for a better judgment.

No information is provided about the optimization. Was it done end to end? For how many epochs? which optimizer? how did the losses from eq. 1 and 4 combine? Is the segmentation decoder loss included in the paper?

Fig 2 and 3: This is the most important figure of the work, and it should be self explanatory. An overview of the pipeline can make the paper simpler to understand, hence increasing its impact and value. Searching for the huge number of notations used in Fig 2 and 3 and arrows entering and exiting from everywhere, confuses the reader. Without reading the text I can never find out what are the arrows entering and exiting the green and purple boxes. Arrows indicate actions or movement of data or order of actions? |D is the support set while D is the decoder, what is x? What are the dashed arrows in Fig 2? I think purpose of the figures is to help the reader understand the text, while in this case text helped me to understand the figures.

In section 3.2, "we do better job on L_intra than L_inter". Are "L_intra & L_inter" losses? seems not to be defined.


Section 2.2: y and x are never defined

**Paper Type:**

methodological development

**Questions To Address In The Rebuttal:**

- Since authors are suggesting a methodology, leaving out the methods is not a good idea
- How did you optimize the model? Is it trained end to end or separately?
- How was the dataset divided into test and train?


----


After the rebuttal, Authors have address a lot of concerns and problems in their paper. I think that's an interesting approach addressing a challenging problem in the field which might be interesting for the target audience.

---

### Official Review · Reviewer_1cqg · 2023-02-03

**Confidence:** 2
**Preliminary Rating:** 3
**Recommendation:** Poster

**Summary:**

The authors present their approach II-CATS to segmenting and tracing neurons from microscopy connectomics data. The authors focus on few-shot learning of intermediate representations. They further introduce two new validation metrics and compare the performance of their method with an FFN baseline and ablations of their model.

**Strengths:**

The general idea seems interesting to me. The paper shows a clear speedup with their method compared to baseline and a improved performance compared to faster methods like watershed. However, I cannot judge the impact of the metrics for this particular application.


**Weaknesses:**

First of all, the paper is not written very clearly. I found it quite hard to follow the explanations.

The overall impact of the paper remains a bit unclear to me as well. Maybe the authors should make this clearer.


**Deanonymize Review:**

no

**Detailed Comments:**

- The overall architecture of the method is not really explained really. Could you give more details here?



**Paper Type:**

methodological development

**Questions To Address In The Rebuttal:**

If the authors could revise the manuscript and improve the clarity of the presentation, it would be much easier to judge the merits of this work. In particular it would be good to motivate and explain the overall architecture and give a bit more care to explain the experiments and the validation.

---

### Official Review · Reviewer_ypEG · 2023-02-06

**Confidence:** 3
**Preliminary Rating:** 3
**Recommendation:** Poster

**Summary:**

The authors propose a novel method (which they call II-CATS) for the tracking and segmentation of neurons in large-scale connectomic reconstruction. Their approach is distinguished by the awareness of intra-cellular and inter-cellular structures simultaneously. The pipeline is validated by making use of new metrics inspired by biological constraints (bERL and nERL), as well as experiments on electron microscopy samples from various datasets, species, and resolutions.

**Strengths:**

* Very complete paper structure in terms of scientific merit. It is appreciated to see ablation studies, implementation details, a detailed description of all modules of the architecture (few shot learner, encoder, decoder, …), and appendixes with all the extra results.
* High-quality figures overall (e.g. figures 2, 3, and 4). They really help the reader here considering the pipeline that is complex.


**Weaknesses:**

Due to the complexity of the pipeline, it is sometimes difficult to follow the “storyline” and ideas of the authors. The writing style seems heavy at times, and a more in-depth grammar check is needed (especially sentence structure). The paper would strongly benefit from some structural changes in the text to help with the flow.

**Deanonymize Review:**

no

**Detailed Comments:**

* How do the datasets compare in terms of each other? What are their characteristics and differences? Why choose these?
In terms of future applications, how do you see other labs adapting your framework for their samples? How transferable would your pipeline be (i.e. would they need to retrain all components of the pipeline, or use the same modules?)?
* It could have been nice to have a table for comparison between datasets (i.e. section 3.1) instead of describing them in the text.
* Recheck the orthograph, grammar, and overall flow of the paper (e.g. paragraph before Table 1).
* Maybe I missed these details in the paper, but how much novelty is there in each of the network modules (few-shot learner, encoder, segmentation decoder, tracker, …)? Is the novelty here mostly in the paradigm (of combining the intra-cellular and inter-cellular information simultaneously) and the 2 new metrics, or are there other novel modules/details?
* There is a lack of future work and limitations discussion in the paper.


**Paper Type:**

both

**Questions To Address In The Rebuttal:**

Same points as above:
* How do the datasets compare in terms of each other? What are their characteristics and differences? Why choose these?
In terms of future applications, how do you see other labs adapting your framework for their samples? How transferable would your pipeline be (i.e. would they need to retrain all components of the pipeline, or use the same modules?)?
* It could have been nice to have a table for comparison between datasets (i.e. section 3.1) instead of describing them in the text.
* Recheck the orthograph, grammar, and overall flow of the paper (e.g. paragraph before Table 1).
* Maybe I missed these details in the paper, but how much novelty is there in each of the network modules (few-shot learner, encoder, segmentation decoder, tracker, …)? Is the novelty here mostly in the paradigm (of combining the intra-cellular and inter-cellular information simultaneously) and the 2 new metrics, or are there other novel modules/details?
* There is a lack of future work and limitations discussion in the paper.

---

### Meta-Review · Area_Chair_LJmi · 2023-02-23

**Recommendation:** Accept (Poster)
**Confidence:** 4

**Metareview:**

All reviewers agreed that the proposed method is an interesting idea and has good values to the MIDL community. All reviewers' criticism and questions were majorly because of the clarity of the original submission. This work could be a nice contribution to MIDL if the authors address all questions (as discussed in the rebuttal) and substantially improve the writing quality in a revised version.